# Efficient Strategies for Computing Euler Number of a 3D Binary Image

Bin Yao [1], Haochen He [1], Shiying Kang [2], Yuyan Chao [3] and Lifeng He [1,4,*]

1 Artificial Intelligence Institute, School of Electronic Information and Artificial Intelligence, Shaanxi University of Science and Technology, Xi'an 710021, China
2 School of Computer Science, Xianyang Normal University, Xianyang 712000, China
3 Faculty of Advanced Business, Nagoya Sangyo University, Aichi 4888711, Japan
4 School of Information Science and Technology, Aichi Prefectural University, Aichi 4801198, Japan
* Correspondence: helifeng@ist.aichi-pu.ac.jp or helf@sust.edu.cn; Tel.: +86-29-86132756

**Abstract:** As an important topological property for a 3D binary image, the Euler number can be computed by finding specific a voxel block with $2 \times 2 \times 2$ voxels, named the voxel pattern, in the image. In this paper, we introduce three strategies for enhancing the efficiency of a voxel-pattern-based Euler number computing algorithm used for 3D binary images. The first strategy is taking advantage of the voxel information acquired during computation to avoid accessing voxels repeatedly. This can reduce the average number of accessed voxels from 8 to 4 for processing a voxel pattern. Therefore, the efficiency of computation will be improved. The second strategy is scanning every two rows and processing two voxel patterns simultaneously in each scan. In this strategy, only three voxels need to be accessed when a voxel pattern is processed. The last strategy is determining the voxel accessing order in the processing voxel pattern and unifying the processing of the voxel patterns that have identical Euler number increments to one group in the computation. Although this strategy can theoretically reduce the average number of voxels accessed from 8 to 4.25 for processing a voxel pattern, it is more efficient than the above two strategies for moderate- and high-density 3D binary images. Experimental results demonstrated that the three algorithms with each of our proposed three strategies exhibit greater efficiency compared to the conventional Euler number computing algorithm based on finding specific voxel patterns in the image.

**Keywords:** 3D image; topological property; Euler number; pattern recognition; computer vision

## 1. Introduction

With the growth of digital images on the web and elsewhere, we need to search, retrieve and classify the images in many applications. Aimed at the problem of image search result organization, Ref. [1] provides a review of the popular methods related to cluster-based image search result organization. By these methods, color images can be categorized by visual features or text features. However, binary images are one of the important images in many image processing systems, and they are often classified by their topological properties. Topological properties of a binary image play a significant role in image recognition, object classification, image segmentation and many other analysis applications. For a binary image, the Euler number remains unchanged despite stretching, flexing or rotating the image, making it a valuable property in many image processing applications, such as medical diagnosis [2], object recognition based on reflectance [3] and crack detection [4]. Numerous algorithms for computing the Euler number of 2D binary images have been used in the past decades, which can be categorized into perimeter-based algorithms [5–7], bit-quad-based algorithms [8,9], run-based algorithms [10,11], graph-based algorithms [12,13] and labeling-based algorithms [14,15].

Recent advancements in image processing and analysis have led to an increase in demand for 3D image processing. The Euler number of 3D images is often used for characterizing complex microstructures [16,17]. For example, it can be used for differentiating the morphologies of graphite; thus, the growth mechanism and the properties of cast iron can be understood precisely.

The definition of Euler number in a 3D binary image is presented by Formula (1).

$$E = C - T + B \tag{1}$$

where $C$ represents the number of connected components, $T$ represents the number of tunnels (or holes), and $B$ represents the number of bubbles (or cavities) in the given image [18,19].

For clarity and convenience, it should be noted that the term "image" henceforth refers exclusively to "binary image".

In addition to this definition, many different kinds of algorithms have been introduced for obtaining the Euler number of 3D images. As introduced above, perimeters and contact perimeters have been used in calculating the Euler number in 2D cases [5–7]. Accordingly, Bribiesca extended it to 3D cases. Ref. [5] presented a perimeter-based algorithm that computes the Euler number of one-voxel-width 3D images according to the perimeters and contact perimeters involved in the connected components of the given image. Akira and Aizawa [20] used an $n \times n$ array of finite-state automata to calculate the numbers of connected components, holes and cavities in the given image, after which the Euler number can be obtained easily by use of Formula (1). Lee and Poston [21] presented an algorithm that smooths the given 3D image and applies theorems of differential geometry and algebraic topology. Moreover, Saha and Chaudhuri proposed an efficient method for obtaining the numbers of connected components, holes and cavities under a specific connectivity relation in [22,23] that can compute the Euler number of the given image using Formula (1). Additionally, Lin et al. introduced an algorithm for computing the Euler number of a 3D image by adding up the number of consecutive voxels, named runs, and adjacent runs existing in the given image [24,25]. This algorithm is extended from 2D cases. Meanwhile, Sánchez-Cruz et al. [26] proposed a new algorithm for computing the Euler number through analyzing the voxelized connected components with cavities and/or tunnels and the relationship between adjacent voxel faces with enclosing surfaces. The algorithm proposed in [26] counts specific voxel patterns, including $1 \times 2 \times 2$, $2 \times 1 \times 2$, $2 \times 2 \times 1$, and $2 \times 2 \times 2$, in the image. In the last few years, two distinctive methods have been presented for calculating the Euler number of a 3D image. Sossa [27] presented a codification-based algorithm that employs the codification of vertices of the foreground voxels in a given image. This algorithm can be thought of as the extension of the algorithm presented in [28]. Moreover, Čomića [29] presented a surface-based formula in computing the Euler number in 3D binary images. This algorithm involves counting only the boundary vertices and faces in the connected components, with the vertex count adjusted for the two adjacency relations.

In [18], Park and Rosenfeld proposed a simple algorithm for computing the Euler number in the given 3D image. This algorithm counts specific $2 \times 2 \times 2$ voxel patterns for the six adjacent cases between voxels. As an extension, Morgenthaler solves the problem of 26-adjacent cases between voxels in [30]. The two algorithms introduced in [18,30] are simple and convenient in implementation.

As mentioned in [20], the judgement of holes is not as easy as that in 2D cases. For example, there are two types of holes. The first type is like the hole in a donut, which allows for string threading. The other type is the hole that can contain jam [21]. Thus, it will raise ambiguities sometimes. Therefore, using a voxel-pattern-based algorithm to compute the Euler number of the given 3D image becomes a relatively good choice. For convenience, this voxel-pattern-based algorithm, proposed by Morgenthaler [30], is denoted as the *VP* algorithm in our paper.

When computing the Euler number of 3D images, the *VP* algorithm must access eight voxels in the corresponding voxel pattern in order to determine whether the current voxel

pattern applies. However, many voxels are accessed repeatedly in the computation. If the repeated accession can be avoided, the algorithm can become more efficient. Based this consideration, three efficiency strategies for enhancing the efficiency of the *VP* algorithm when computing the Euler number of the given 3D image are introduced in this paper. The first strategy is taking advantage of the information of voxels acquired from the previous processed $2 \times 2 \times 2$ voxel pattern to avoid accessing voxels repeatedly. Then, it is possible to reduce the average number of accessed voxels required for processing a voxel pattern from 8 to 4. Thus, the computing efficiency can be improved. We implement this strategy using state transition; thus, the algorithm with this strategy is denoted as the *ST* algorithm in this paper. The second strategy is scanning every two rows and processing two voxel patterns simultaneously. By this strategy, only three voxels need to be accessed for processing a voxel pattern in our proposed algorithm. Because more rows are scanned simultaneously in this strategy, the algorithm with this strategy is denoted as the *MR* algorithm in this paper. The last strategy focuses on the accessing order when processing a voxel pattern. We change the accessing order of voxels when processing voxel patterns, and at the same time, the voxel patterns with identical Euler number increments are combined to the same group for processing. Theoretically, when we process a voxel pattern by this strategy, the reduction of the average number of accessed voxels would be decreased from 8 to 4.25, but it is more efficient for moderate- and high-density images. In this strategy, we change the voxel accessing orders when we process a voxel pattern; thus, the algorithm with this strategy is denoted as the *CO* algorithm in this paper. Experimental results demonstrated that the algorithms with our strategies exhibit greater efficiency in comparison to the *VP* algorithm.

The organization of the paper is as follows: we review the conventional voxel-pattern-based Euler number computing algorithm in Section 2 and present our strategies in Section 3. In Section 4, experiments are conducted on different resolutions and different densities of noise images for evaluating the efficiency of different strategies in comparison to the *VP* algorithm. We discuss the algorithms in Section 5, and we present concluding remarks in the last section.

## 2. Reviews of Conventional Voxel-Pattern-Based Euler Number Computing Algorithm for a 3D Image

A 3D image can be described as an array of volume elements, or voxels. For the binary case, a 3D image can be considered as a relation or function $f: \Sigma \to \{0, 1\}$, which maps from $\Sigma$ to the set $\{0, 1\}$. To determine the adjacency between voxels in the image, a pair of voxels $P = (a_1, a_2, a_3)$ and $Q = (b_1, b_2, b_3)$ are considered to be 6-adjacent if their Manhattan distance is equal to 1, i.e., $|a_1 - b_1| + |a_2 - b_2| + |a_3 - b_3| = 1$. Alternatively, $P$ and $Q$ are considered to be 26-adjacent if their Chebyshev distance is equal to 1, i.e., $\max(|a_1 - b_1|, |a_2 - b_2|, |a_3 - b_3|) = 1$. As illustrated in Figure 1, $p_1, p_3, p_5, p_7, p_{17}$ and $p_{26}$ are 6-adjacent to voxel $p$, and voxels $p_1, p_2, \ldots, p_{26}$ are 26-adjacent to voxel $p$.

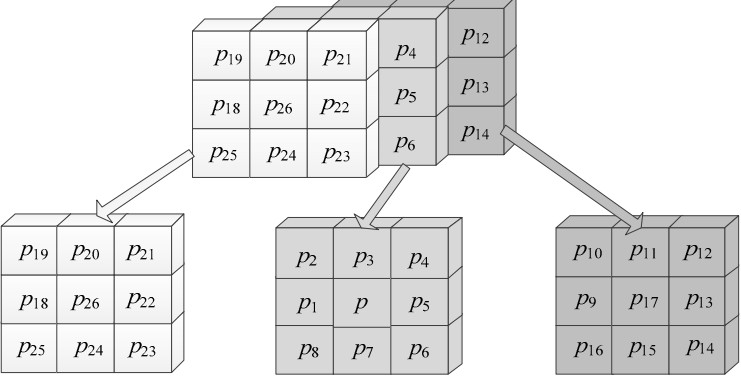

**Figure 1.** Illustration for explaining different adjacency among voxels.

We assume that the foreground voxels are expressed by 1 and the background voxels are expressed by 0 in a 3D image. Furthermore, all voxels on the border of the given image are assumed to be background voxels, which is the same as in most image processing algorithms. Additionally, only 26-adjacent between foreground voxels are considered in this paper.

In order to calculate the Euler number, specific configurations of $2 \times 2 \times 2$ voxel patterns need to be found in the given image in the *VP* algorithm, as shown in Figure 2. Let #[*x*] ($1 \leq x \leq 22$) represents the number of occurrences that the voxel pattern *i* found in the given image, and then we can compute the Euler number according to Formula (2) [30].

$$E = \Psi_1 - \Psi_2 + \Psi_3 - \Psi_4 + \Psi_5 - \Psi_6 + \Psi_7 - \Psi_8 \qquad (2)$$

where

$\Psi_1 = \#[1];$
$\Psi_2 = \#[2] + \#[3] + \#[4];$
$\Psi_3 = \#[5] + \#[6] + \#[7];$
$\Psi_4 = \#[8] + \#[9] + \#[10] + \#[11] + \#[12] + \#[13] + \#[14];$
$\Psi_5 = \#[15] + \#[16] + \#[17];$
$\Psi_6 = \#[18] + \#[19] + \#[20];$
$\Psi_7 = \#[21];$
$\Psi_8 = \#[22].$

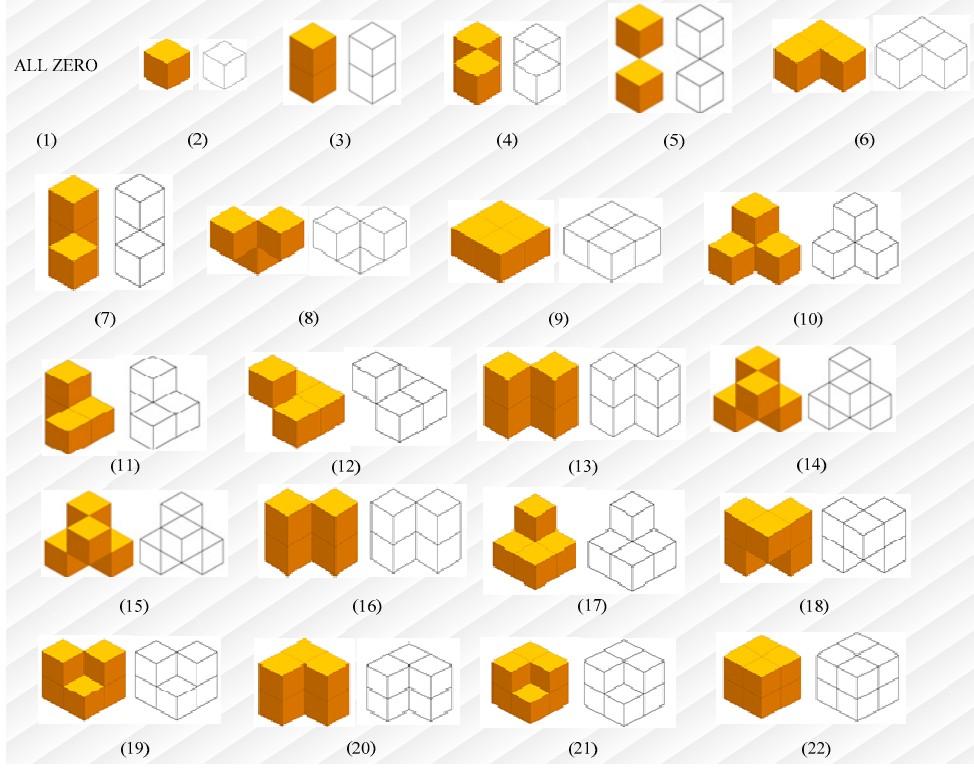

**Figure 2.** Different voxel patterns found in the algorithm proposed in [30].

In is known that a $2 \times 2 \times 2$ voxel pattern consists of eight voxels, and each of them is either a foreground voxel or a background voxel. Accordingly, there will be 256 different types of configurations formed by the eight voxels, theoretically. To determine a voxel pattern's configuration as found in the image, it is necessary to check it from all orientations. In practice, it is difficult to determine the voxel patterns' configurations occurring in the image from all orientations. To address this issue, Morgenthaler summed up the Euler number increments of 256 different types of voxel patterns in the *VP* algorithm and listed their increments (see Table 1). The table lists the Euler number increment $\Delta E$ of voxel patterns that are not zero according to an index generated by the values of voxels in the

voxel pattern. For instance, for the voxel pattern presented in Figure 3, if the values of voxels *a*, *b*, *c*, *d*, *e*, *f*, *g* and *h* are 1, 0, 1, 1, 1, 1, 1 and 1, respectively, the index should be 10111111, and from Table 1, it can be concluded that the Euler number increment of the current voxel pattern will be 1. In other words, once this pattern is found in the image, the Euler number of the image should be increased by 1. The Euler number increments of the other voxel patterns should be decided using a similar method.

**Table 1.** Voxel patterns' indexes and their Euler number increments.

| Index of Voxel Patterns | $\Delta E$ | Index of Voxel Patterns | $\Delta E$ | Index of Voxel Patterns | $\Delta E$ |
|---|---|---|---|---|---|
| 00000010 | 1 | 00001001 | −1 | 00001011 | −1 |
| 00011000 | −1 | 00011001 | −1 | 00011010 | −1 |
| 00011011 | −1 | 00100001 | −1 | 00100011 | −1 |
| 00100100 | −1 | 00100101 | −1 | 00100110 | −1 |
| 00100111 | −1 | 00101000 | −1 | 00101001 | −2 |
| 00101010 | −1 | 00101011 | −2 | 00101100 | −1 |
| 00101101 | −1 | 00101110 | −1 | 00101111 | −1 |
| 00111000 | −1 | 00111001 | −1 | 00111010 | −1 |
| 00111011 | −1 | 10000001 | −1 | 10000011 | −1 |
| 10001001 | −1 | 10001011 | −1 | 10010100 | 1 |
| 10010101 | 1 | 10010110 | 1 | 10010111 | 1 |
| 10011100 | 1 | 10011101 | 1 | 10011110 | 1 |
| 10011111 | 1 | 10100001 | −1 | 10100011 | −1 |
| 10101001 | −1 | 10101011 | −1 | 10110100 | 1 |
| 10110101 | 1 | 10110110 | 1 | 10110111 | 1 |
| 10111100 | 1 | 10111101 | 1 | 10111110 | 1 |
| 10111111 | 1 | Others | 0 | | |

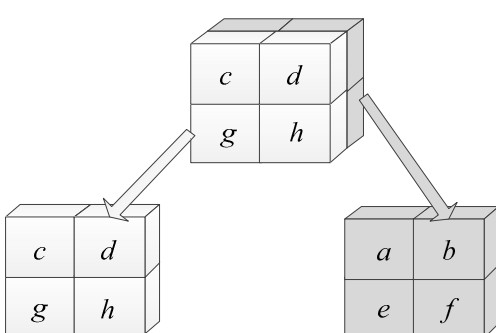

**Figure 3.** Distribution of voxels in a voxel pattern.

Although 256 different types voxel patterns can be formed by a 2 × 2 × 2 voxel block, as listed in Table 1, there are only 49 types of voxel patterns having non-zero Euler number increments. In other words, the increment of all the other 207 types of the voxel patterns is 0; thus, they will not have an influence on the Euler number of the image [30]. Among the voxel patterns which will have influences on the Euler number, 30 types of voxel patterns' increments are −1, 17 types of voxel patterns' increments are 1 and the last 2 types of voxel patterns' increments are −2.

The *VP* algorithm is practicable in implementation. According to this algorithm, when computing the Euler number, it is necessary for us to access voxels in the image one by one, confirm the corresponding voxel patterns' indexes and consult Table 1 for their Euler number increments. Once all voxels have been processed and all voxel patterns' Euler number increments have been determined, we can obtain the Euler number of the given image easily. It is obvious that for finding the voxel patterns that will have influences on the Euler number, all voxel patterns' Euler number increments have to be determined, and all the eight voxels have to be accessed in each voxel pattern. Therefore 8 × *X* × *Y* × *Z* voxel

accessions are required for computing the Euler number of an image with a resolution of $X \times Y \times Z$ voxels in the *VP* algorithm.

## 3. Our Proposed Strategies for Improving the *VP* Algorithm

As introduced above, while calculating the Euler number, for each voxel found in the raster scan, the *VP* algorithm will access all voxels in the corresponding voxel pattern when confirming whether the voxel pattern should be applied. As a matter of fact, plenty of voxels are accessed repeatedly in the computing procedure. In this section, we will introduce some strategies for reducing the number of voxel accessions while processing a voxel pattern.

### 3.1. Strategy of State Transition

In the *VP* algorithm, many voxels would be accessed repeatedly. For example, as shown in Figure 4, when processing voxel $a_1$, it is required to access eight voxels in the corresponding voxel pattern $\{a_1, b_1, i_1, j_1, a_2, b_2, i_2, j_2\}$ and confirm its index of the pattern. After doing this, it goes on to process the next voxel, i.e., voxel $b_1$, and the same method is used as in processing voxel $a_1$: eight voxels in the corresponding voxel pattern $\{b_1, c_1, j_1, k_1, b_2, c_2, j_2, k_2\}$ will be accessed, where the voxels $b_1, j_1, b_2$ and $j_2$ have already been accessed while processing the previous voxel $a_1$. To prevent the repeated accesses, we can utilize the information of voxels $b_1, j_1, b_2$ and $j_2$ obtained during processing the voxel $a_1$.

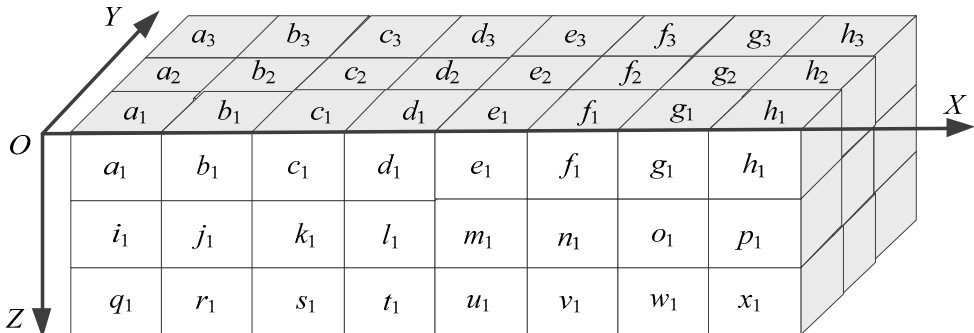

**Figure 4.** Illustration of the repeated accessing of voxels existing in the *VP* algorithm.

In order to reduce the accession of voxels considered above, we construct sixteen different states for a quad, which are denoted as $St_0, St_1, \ldots, St_{15}$, as shown in Figure 5. In this way, the left quad (four voxels) in the voxel pattern being processed can be replaced by the corresponding state, and thus, by utilizing transitions among the states, we only need to access the four voxels in the right quad in the current voxel pattern. Then, we describe the strategy in detail.

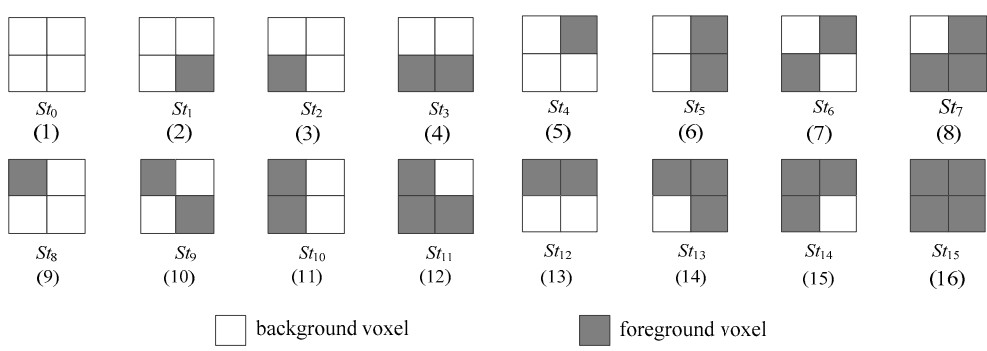

**Figure 5.** Sixteen states defined in the strategy of state transition.

Because all voxels on the border in a 3D image are considered to be background voxels, the left quad of the first voxel pattern will be the state $St_0$. Thus, we will begin

our processing based on the state $St_0$ and access the other four voxels in the right quad for determining the index of the first voxel pattern of the image. For example, when the left quad of the current voxel pattern $\{a_1, b_1, i_1, j_1, a_2, b_2, i_2, j_2\}$, i.e., $\{a_2, a_1, i_2, i_1\}$, shown in Figure 4, is the case $St_0$, we only need to access the voxels in the right quad $\{b_2, b_1, j_2, j_1\}$ for determining the index of the voxel pattern. After doing this, we proceed to process the following voxel pattern $\{b_1, c_1, j_1, k_1, b_2, c_2, j_2, k_2\}$, where the left quad is $\{b_2, b_1, j_2, j_1\}$, which will be a certain state as listed in Figure 5 according to the values of voxels $b_2$, $b_1$, $j_2$ and $j_1$. Therefore, we only need to access the four voxels in the right quad $\{c_2, c_1, k_2, k_1\}$.

Then, let us consider, when the left quad $\{b_2, b_1, j_2, j_1\}$ is, for example, the state $St_7$, i.e., $\{0, 1, 1, 1\}$, how to process the voxel pattern $\{b_1, c_1, j_1, k_1, b_2, c_2, j_2, k_2\}$. We first check the voxels in the right quad $\{c_2, c_1, k_2, k_1\}$ to confirm its state. For each possible state $St_i$ ($0 \leq i \leq 15$), we go a step further to determine the Euler number increment of the current voxel pattern by consulting Table 1. Then, let us consider some cases.

(1) If the values of voxels $c_2$, $c_1$, $k_2$ and $k_1$ are 0, as shown in Figure 4, the voxel pattern being processed will be $\{0, 0, 1, 0, 1, 0, 1, 0\}$. According to Table 1, we know that the current voxel pattern needs to be involved in the computation, and its Euler number increment is $-1$. Thus, the Euler number of the given image will decrease by 1. After doing this, we proceed to process the next one in the image. Because the values of voxels $c_2$, $c_1$, $k_2$ and $k_1$ are 0, we will process the next one from state $St_0$.

(2) If the values of voxels $c_2$, $c_1$, $k_2$ and $k_1$ are 0, 0, 0 and 1, respectively, the voxel pattern being processed will be $\{0, 0, 1, 0, 1, 0, 1, 1\}$. According to Table 1, we know that the current voxel pattern needs to be involved in the computation, and its Euler number increment is $-2$. Therefore, the Euler number will decrease by 2. After doing this, we proceed to process the next one in the image. Because the values of voxels $c_2$, $c_1$, $k_2$ and $k_1$ are 0, 0, 0 and 1, we will process the next one from state $St_1$.

(3) If the values of voxels $c_2$, $c_1$, $k_2$ and $k_1$ are 0, 1, 1 and 0, respectively, the voxel pattern being processed will be $\{0, 0, 1, 1, 1, 1, 1, 0\}$. According to Table 1, we know that the current voxel pattern does not need to be involved in the computation on account of its zero Euler number increment, and this voxel pattern will not have an influence on the Euler number of the image. Then we proceed to process the next voxel pattern. Because the values of voxels $c_2$, $c_1$, $k_2$ and $k_1$ are 0, 1, 1 and 0, we will process the next one from state $St_6$.

Similar procedures can be applied to process other cases. The processing results are presented in Table 2.

**Table 2.** The Euler number increment of a voxel pattern whose left quad is $St_7$.

| The State of the Right Quad | Index | $\Delta E$ | The State of the Right Quad | Index | $\Delta E$ |
|---|---|---|---|---|---|
| $St_0$ | 00101010 | $-1$ | $St_8$ | 01101010 | 0 |
| $St_1$ | 00101011 | $-2$ | $St_9$ | 01101011 | 0 |
| $St_2$ | 00101110 | $-1$ | $St_{10}$ | 01101110 | 0 |
| $St_3$ | 00101111 | $-1$ | $St_{11}$ | 01101111 | 0 |
| $St_4$ | 00111010 | $-1$ | $St_{12}$ | 01111010 | 0 |
| $St_5$ | 00111011 | $-2$ | $St_{13}$ | 01111011 | 0 |
| $St_6$ | 00111110 | 0 | $St_{14}$ | 01111110 | 0 |
| $St_7$ | 00111111 | 0 | $St_{15}$ | 01111111 | 0 |

For each of the other states of the left quad in the voxel pattern, we can process the voxel pattern in a similar way. This procedure will be executed recursively until we finish processing all voxel patterns. Thus, we acquire the indexes and the same increments of all voxel patterns as in the *VP* algorithm. Consequently, the Euler number of the given image can be obtained according to the increments of different voxel patterns easily.

In this strategy, taking advantage of the voxel information acquired while processing the previous voxels, the number of accessed voxels decreased from 8 to 4 while processing each voxel pattern, which is half of that in the *VP* algorithm. Thus, it results in more

efficient computing. For convenience, the state-transition-based algorithm described above is denoted as the *ST* algorithm in this paper.

### 3.2. Scanning Every Two Rows and Processing Two Voxel Patterns Simultaneously

As introduced above, when processing a voxel pattern, the *ST* algorithm can avoid accessing the four voxels that have been accessed by making use of the voxel information acquired while processing the previous voxels. Nevertheless, many voxels are still accessed repeatedly in the *ST* algorithm. As shown in Figure 4, for processing the voxel $a_1$ in the first row, eight voxels in the voxel pattern $\{a_1, b_1, i_1, j_1, a_2, b_2, i_2, j_2\}$ need to be accessed. Then, when we process the voxel $i_1$ in the next row, another eight voxels in the voxel pattern $\{i_1, j_1, q_1, r_1, i_2, j_2, q_2, r_2\}$ need to be accessed. Obviously, voxels $i_1, j_1, i_2$ and $j_2$ will be accessed repeatedly.

We can avoid accessing such voxels repeatedly if we scan every two rows simultaneously, and for each voxel being processed, we access the involved twelve voxels to determine the indexes of two voxel patterns simultaneously. For example, when processing voxel $a_1$ in Figure 4, we access twelve voxels simultaneously, i.e., $a_1, b_1, i_1, j_1, q_1, r_1, a_2, b_2, i_2, j_2, q_2$ and $r_2$, to determine the indexes of two voxel patterns $\{a_1, b_1, i_1, j_1, a_2, b_2, i_2, j_2\}$ and $\{i_1, j_1, q_1, r_1, i_2, j_2, q_2, r_2\}$ simultaneously. For convenience, we represent the two voxel patterns as $VP_1$ and $VP_2$, respectively.

Furthermore, when we access twelve voxels to process two voxel patterns simultaneously, similar to that in the *ST* algorithm, by use of states transitions, we only need to access the voxels $b_1, j_1, r_1, b_2, j_2$, and $r_2$. Obviously, there are $2^6 = 64$ states, as shown in Figure 6, to be defined.

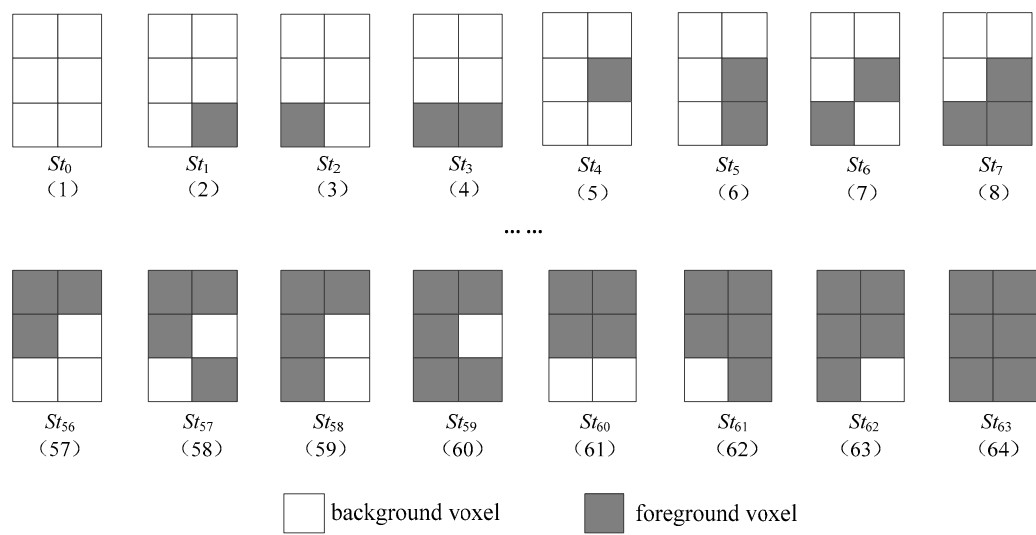

**Figure 6.** Sixty-four states defined in processing three rows simultaneously.

Similar to that in the *ST* algorithm, because all voxels on the border of a 3D image are background voxels, our computation should begin from state $St_0$, as shown in Figure 6(1), and access the other six voxels to determine the indexes of voxel patterns. As shown in Figure 4, when we process the first voxel $a_1$, because voxels $a_2, a_1, i_2, i_1, q_2$ and $q_1$ on the border are background voxels and the corresponding state is the state $St_0$, we only need to access voxels $b_2, b_1, j_2, j_1, r_2$ and $r_1$ to determine the indexes of voxel patterns $\{a_2, a_1, i_2, i_1, b_2, b_1, j_2, j_1\}$ and $\{i_2, i_1, q_2, q_1, j_2, j_1, r_2, r_1\}$, respectively. After doing this, taking advantage of the state determined by the values of voxels $b_2, b_1, j_2, j_1, r_2$ and $r_1$, we go on to access the next six voxels $c_2, c_1, k_2, k_1, s_2$ and $s_1$ to confirm the indexes of voxel patterns $\{b_2, b_1, j_2, j_1, c_2, c_1, k_2, k_1\}$ and $\{j_2, j_1, r_2, r_1, k_2, k_1, s_2, s_1\}$, respectively.

Then, let us consider, when the left six voxels $\{b_2, b_1, j_2, j_1, r_2, r_1\}$ are, for example, the state $St_7$, i.e., $\{0, 0, 0, 1, 1, 1\}$, how to process the voxel patterns $\{b_1, c_1, j_1, k_1, b_2, c_2, j_2, k_2\}$ and $\{j_2, j_1, r_2, r_1, k_2, k_1, s_2, s_1\}$. We first check the six voxels to the right of the twelve voxels,

i.e., $\{c_2, c_1, k_2, k_1, s_2, s_1\}$ to confirm its state $St_i$ $(0 \leq i \leq 63)$. Then, by using of two voxel patterns derived from the twelve voxels, we further confirm the Euler number increment of the corresponding voxel pattern by referring to Table 1. We use the following cases for explanation.

(1) If the values of voxels $c_2$, $c_1$, $k_2$, $k_1$, $s_2$ and $s_1$ are 0, 0, 0, 0, 0 and 0, respectively, then $VP_1$ will be $\{0, 0, 0, 0, 0, 0, 1, 0\}$ and $VP_2$ will be $\{0, 0, 1, 0, 1, 0, 1, 0\}$. According to Table 1, we know that $VP_1$ should be involved in the computation, and its Euler number increment is 1. At the same time, $VP_2$ should be also involved, and its Euler number increment is $-1$. Then we proceed to process the next voxel pattern. Because the values of voxels $c_2$, $c_1$, $k_2$, $k_1$, $s_2$ and $s_1$ are 0, 0, 0, 0, 0 and 0, we will process the next one from state $St_0$.

(2) If the values of voxels $c_2$, $c_1$, $k_2$, $k_1$, $s_2$ and $s_1$ are 0, 0, 0, 0, 1 and 1, respectively, then $VP_1$ will be $\{0, 0, 0, 0, 0, 0, 1, 0\}$ and $VP_2$ will be $\{0, 0, 1, 0, 1, 1, 1, 1\}$. According to Table 1, we know that $VP_1$ should be involved in the computation, and its Euler number increment is 1. However, $VP_2$ should not be involved because its Euler number increment is 0. Then we proceed to process the next voxel pattern. Because the values of voxels $c_2$, $c_1$, $k_2$, $k_1$, $s_2$ and $s_1$ are 0, 0, 0, 0, 1 and 1, we will process the next one from state $St_3$.

(3) If the values of voxels $c_2$, $c_1$, $k_2$, $k_1$, $s_2$ and $s_1$ are 1, 1, 1, 1, 1 and 1, respectively, then $VP_1$ will be $\{0, 1, 0, 1, 0, 1, 1, 1\}$ and $VP_2$ will be $\{0, 1, 1, 1, 1, 1, 1, 1\}$. According to Table 1, we know that both $VP_1$ and $VP_2$ should not be involved in the computation because their Euler number increments are 0. Then, we proceed to process the next voxel pattern. Because the values of voxels $c_2$, $c_1$, $k_2$, $k_1$, $s_2$ and $s_1$ are 1, 1, 1, 1, 1 and 1, we will process the next one from state $St_{63}$.

Similar procedures can be applied to process other cases. The processing results are presented in Table 3.

**Table 3.** The Euler number increment of a voxel pattern whose left six voxels are $St_7$.

| State of Right Voxels | Voxel Pattern | Index | $\Delta E$ | State of Right Voxels | Voxel Pattern | Index | $\Delta E$ |
|---|---|---|---|---|---|---|---|
| $St_0$ | $VP_1$ | 00000010 | 1 | $St_1$ | $VP_1$ | 00000010 | 1 |
| | $VP_2$ | 00101010 | $-1$ | | $VP_2$ | 00101011 | $-2$ |
| $St_2$ | $VP_1$ | 00000010 | 1 | $St_3$ | $VP_1$ | 00000010 | 1 |
| | $VP_2$ | 00101110 | $-1$ | | $VP_2$ | 00101111 | $-1$ |
| $St_4$ | $VP_1$ | 00000011 | 0 | $St_5$ | $VP_1$ | 00000011 | 0 |
| | $VP_2$ | 00111010 | $-1$ | | $VP_2$ | 00111011 | $-1$ |
| $St_6$ | $VP_1$ | 00000011 | 0 | $St_7$ | $VP_1$ | 00000011 | 0 |
| | $VP_2$ | 00111110 | 0 | | $VP_2$ | 00111111 | 0 |
| | | … … | | | | … … | |
| | | … … | | | | … … | |
| $St_{56}$ | $VP_1$ | 01010110 | 0 | $St_{57}$ | $VP_1$ | 01010110 | 0 |
| | $VP_2$ | 01101010 | 0 | | $VP_2$ | 01101011 | 0 |
| $St_{58}$ | $VP_1$ | 01010110 | 0 | $St_{59}$ | $VP_1$ | 01010110 | 0 |
| | $VP_2$ | 01101110 | 0 | | $VP_2$ | 01101111 | 0 |
| $St_{60}$ | $VP_1$ | 01010111 | 0 | $St_{61}$ | $VP_1$ | 01010111 | 0 |
| | $VP_2$ | 01111010 | 0 | | $VP_2$ | 01111011 | 0 |
| $St_{62}$ | $VP_1$ | 01010111 | 0 | $St_{63}$ | $VP_1$ | 01010111 | 0 |
| | $VP_2$ | 01111110 | 0 | | $VP_2$ | 01111111 | 0 |

After all voxel patterns are processed in the image, we can acquire the same increments of all voxel patterns as in the $VP$ algorithm. Thus, the Euler number of the image can be obtained easily on the basis of increments of different voxel patterns.

In this strategy, we scan every two image rows, and for each voxel in processing, we access six voxels to process two voxel patterns simultaneously. That is to say, for processing a voxel pattern, the average number of voxels needing to be accessed will be $6/2 = 3$, which is fewer than that in the $ST$ algorithm; thus, it might lead to more efficient processing.

For convenience, this multi-row-scan-based algorithm is denoted as the *MR* algorithm in this paper.

### 3.3. Strategy of Changing the Order of Accessing Voxels and Combining Similar Voxel Patterns

We continue to analyze the information in Table 1. It is not difficult to find that some voxel patterns' indexes are consecutive, as listed in Table 4. Moreover, we can observe that these voxel patterns with consecutive indexes have something in common, i.e., these voxel patterns have the same Euler number increments. Accordingly, we can divide these voxel patterns into some groups. For example, for each voxel pattern indexed by 00011000, 00011001, 00011010 or 00011011 occurring in the processing, the Euler number will decrease by 1. On the other hand, it can be noticed that these consecutive indexes are different in the last two bits, which vary from 00 to 11.

**Table 4.** Euler number increments of voxel patterns with consecutive indexes.

| Indexes of Voxel Patterns | $\Delta E$ | Indexes of Voxel Patterns | $\Delta E$ |
|---|---|---|---|
| 00011000–00011011 | $-1$ | 10010100–10010111 | 1 |
| 00100100–00100111 | $-1$ | 10011100–10011111 | 1 |
| 00101100–00101111 | $-1$ | 10110100–10110111 | 1 |
| 00111000–00111011 | $-1$ | 10111100–10111111 | 1 |

Then, if the voxel patterns in each group are considered to be a voxel pattern cluster, we can represent the cluster by the first six voxels. In other words, the Euler number increments of the voxel patterns in these clusters can be acquired by accessing the first six voxels. For example, for the processing voxel pattern, if the values of the first six voxels *a*, *b*, *c*, *d*, *e* and *f* are 1, 0, 1, 1, 1 and 1, respectively, according to Table 4, we can deduce that the Euler number increment of the current voxel pattern is 1. In this case, we can acquire its Euler number increment without accessing voxel *g* and voxel *h*. Obviously, for processing a voxel pattern listed in Table 4, only six voxels need to be accessed.

By taking into account the aforementioned consideration, we can apply the same approach to the other groups of voxel patterns listed in Table 4. As shown in Table 4, 32 types of voxel patterns' Euler number increments can be determined by the voxel information of the six voxels. Thus, this strategy will result in a more efficient computation.

We conducted further analysis on the voxel pattern indexes that have influences on the Euler number of the given image. An important observation was that the voxel located at the position "*b*" in each of these indexes is always a background voxel. Based on this observation, when we process a voxel pattern, we should focus on the voxel at the position "*b*" first. When the voxel "*b*" is found to be a background voxel, then we have to access other voxels in the voxel pattern for determining the increment. When the voxel "*b*" is found to be a foreground voxel, no voxels need to be accessed further, because the current voxel pattern Euler number increment must be 0. By this means, while processing a voxel pattern with a foreground voxel at the position "*b*", only one voxel needs to be accessed for determining its Euler number increment.

In implementation, the two methods described above can be combined for improving the *VP* algorithm, and we can process a voxel pattern according to the following steps.

When processing the voxel pattern {*a*, *b*, *c*, *d*, *e*, *f*, *g*, *h*}, we first access the voxel "*b*" in its index.

(1) If the voxel "*b*" is a foreground voxel, it can be confirmed that the current voxel pattern does not need to be involved in the computation because its Euler number increment must be 0. Subsequently, we proceed to process the next one. In this case, only one voxel needs to be accessed for determining these voxel patterns' Euler number increments.

(2) If the voxel "*b*" is a background voxel, we must conduct a sequential accessing of the voxels *a*, *c*, *d*, *e* and *f* of the current voxel pattern. If the values of voxels *a*, *c*, *d*, *e* and *f* are equal to one of the value groups {0, 0, 1, 1, 0}, {0, 1, 0, 0, 1}, {0, 1, 0, 1, 1}, {0, 1, 1, 1, 0}, {1,

0, 1, 0, 1}, {1, 0, 1, 1, 1}, {1, 1, 1, 0, 1} or {1, 1, 1, 1, 1}, following the list in Table 4, only six voxels need to be accessed for determining these voxel patterns' Euler number increments.

(3) Otherwise, the two remaining voxels need to be accessed for determining the Euler number increment of the current voxel pattern. In this case, eight voxels need to be accessed for determining the Euler number increment.

By utilizing the above steps, we can process all voxel patterns in the given image and acquire the increments of all voxel patterns. Accordingly, the Euler number of the given image can be obtained easily. In this strategy, we change the accessing order of voxel patterns. For convenience, this changing-order-based algorithm is denoted as the *CO* algorithm in this paper.

## 4. Experimental Results

In this section, experiments are conducted on different resolutions of noise images for verifying the efficiency of different strategies in comparison to the *VP* algorithm. The performance of the *ST* algorithm, the *MR* algorithm and the *CO* algorithm will be compared with the *VP* algorithm. The efficiency of the algorithms is evaluated by their execution time in processing the same 3D images. All of the algorithms used for comparison were implemented in C language. The experimental environment and the experimental platform are presented in Table 5.

**Table 5.** The experimental setup and configurations of hardware and software.

| Name | Version |
|---|---|
| Processor | Intel Core i7-6770 |
| Frequency | 3.20 GHz |
| Memory | 8 GB |
| Operating System | Ubuntu Linux 20.04.1 |
| GCC Compiler | 4.6.1 |

Because noise images have random voxel distribution and complex connectivity among voxels, they are conducive to evaluating the performance of the algorithms. In our experiments, five resolutions ($32 \times 32 \times 32, 64 \times 64 \times 64, 128 \times 128 \times 128, 256 \times 256 \times 256, 512 \times 512 \times 512$ voxels) of 3D noise images were tested. For each resolution, 41 noise images were generated by thresholding of the images containing uniform random noise with 41 different values from 0 to 1000 in steps of 25. We presented nine cross sections of 3D noise images with densities of 0.1 to 0.9 in steps of 0.1 in Figure 7. In order to make the experimental results more accurate, we repeated the test 1000 times and obtained the experimental results by averaging of the execution time of the compared algorithms.

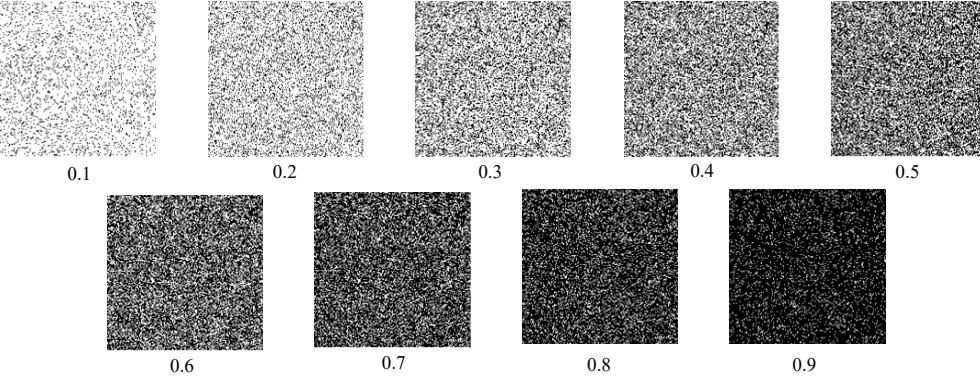

**Figure 7.** Crossing sections of noise images with different densities.

### 4.1. Execution Time versus the Number of Voxels in the Image

In this experiment, a total of 205 noise images with different resolutions were tested for evaluating the execution time versus the number of voxels for the compared algorithms.

The experimental results are given in Figure 8. From Figure 8, it is found that for both the maximum execution time and the average execution time, all algorithms in comparison hold good linear features versus the number of voxels in the image. Furthermore, the algorithms with our proposed strategies perform more efficiently than the *VP* algorithm. Among the algorithms with our proposed strategies, the *CO* algorithm is the most efficient.

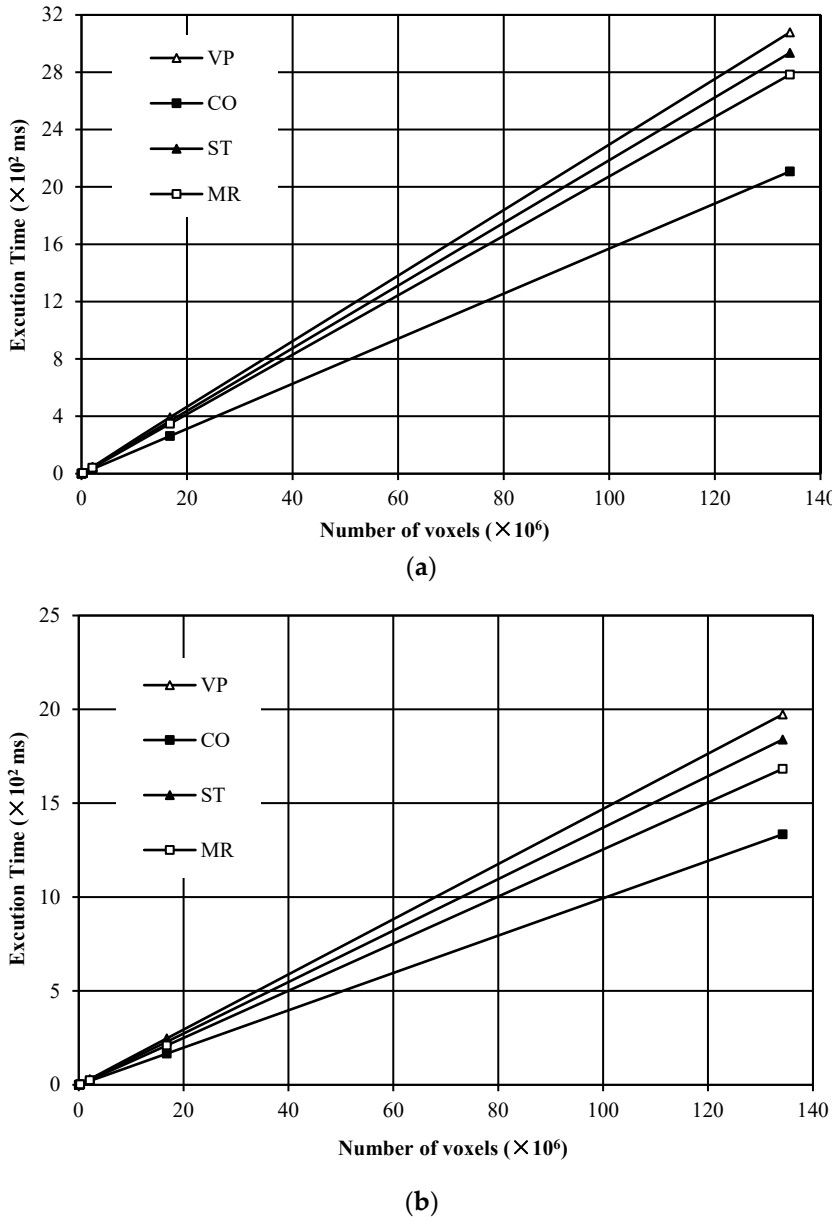

(**a**)

(**b**)

**Figure 8.** Execution time versus the number of voxels in the image: (**a**) the maximum execution time; (**b**) the average execution time.

### 4.2. Comparison of the Number of Accessed Voxels in the Compared Algorithms

In this experiment, 41 noise images with a resolution of $64 \times 64 \times 64$ voxels were tested for comparing the number of accessed voxels among the compared algorithms. The experimental results are shown in Figure 9. From Figure 9, it can be found that when all voxels are background voxels in the image, the number of accessed voxels in the *VP* algorithm is the same as that in the *CO* algorithm. With the increased densities of the foreground voxels in the images, the number of voxels needing to be accessed in the *CO* algorithm gradually becomes fewer than that in the *VP* algorithm. When all voxels are foreground voxels in the image, the number of voxels needing to be accessed in the *CO*

algorithm is less than 1/10 of that in the *VP* algorithm. Moreover, for all tested images, the *VP* algorithm needs to access the same number of voxels. Similarly, the *ST* algorithm needs to access the same number of voxels for processing any image, but the number of accessed voxels is about half of that in the *VP* algorithm. As to the *MR* algorithm, for processing any image, it also needs to access the same number of voxels, which are fewer than that in the *ST* algorithm. This is consistent with our analysis.

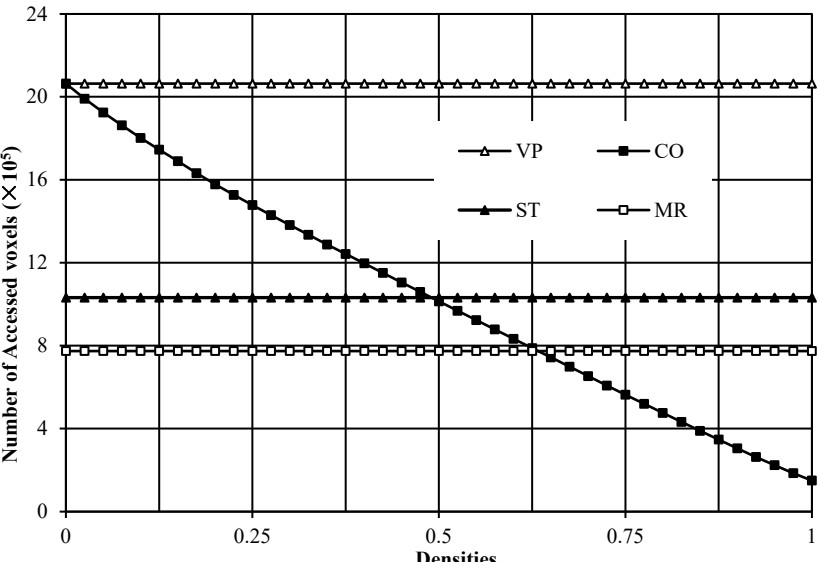

**Figure 9.** Number of accessed voxels in the compared algorithms.

### 4.3. Execution Time versus Image Densities

In this experiment, 41 noise images with a resolution of $512 \times 512 \times 512$ voxels are employed for verifying the execution time versus the densities of foreground voxels in the image. Figure 10 presents the trend of execution time of the compared algorithms. From Figure 10, it is found that the algorithms with our presented strategies are more efficient than the *VP* algorithm for almost all images.

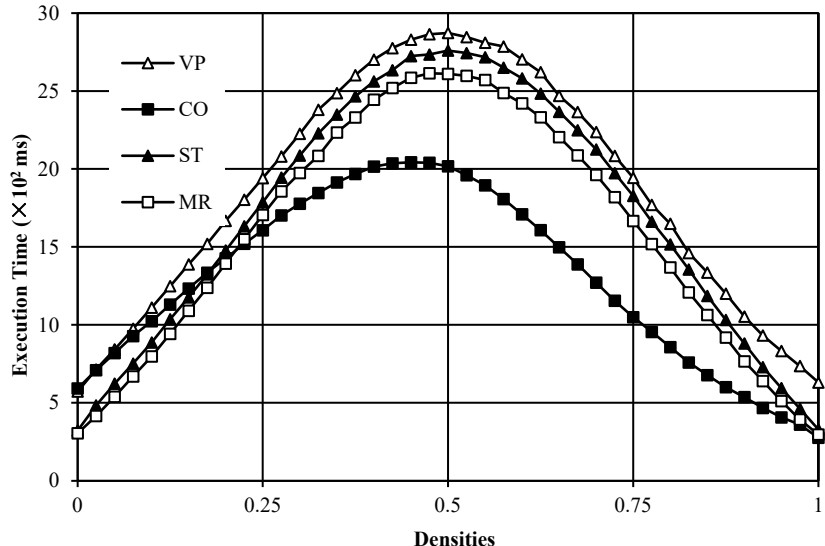

**Figure 10.** Execution time versus image densities of foreground voxels.

Furthermore, as depicted above, for processing a voxel pattern, the number of voxels accessed in the *ST* algorithm and the *MR* algorithm is fewer than that in the *VP* algorithm. Therefore, the *ST* algorithm and the *MR* algorithm outperformed the *VP* algorithm. How-

ever, although the *ST* algorithm only accesses half of the number of voxels that the *VP* algorithm does, when we calculate the Euler number of a 3D image, the execution time of the *ST* algorithm is not half of that of the *VP* algorithm. Moreover, for all tested images, although the *MR* algorithm performs more efficiently than the *ST* algorithm, the difference between the two algorithms is minimal.

As to the *CO* algorithm, it is much more efficient than the other compared algorithms, in particular for those images whose densities of foreground voxels vary from 0.4 to 0.8. However, it is less efficient than the *ST* algorithm and the *MR* algorithm when the densities of foreground voxels are lower than 0.2.

## 5. Discussion

In this paper, we study the voxel-pattern-based Euler number computing algorithms. The *VP* algorithm proposed by Morgenthaler needs to access eight voxels in the corresponding voxel pattern for processing a voxel of the image. Thus, for a 3D image with a resolution of $X \times Y \times Z$ voxels, $8 \times X \times Y \times Z$ voxels need to be accessed for computing the Euler number. Taking advantage of the voxel information acquired while processing the previous voxel patterns, the number of voxels needing to be accessed can decrease to four for processing a voxel pattern in the *ST* algorithm. The *MR* algorithm scans every two rows and processes voxel patterns two by two; only three voxels need to be accessed for processing a voxel pattern.

In fact, we could scan every three rows and process voxel patterns three by three. In this case, taking advantage of the voxel information acquired during processing of the previous voxel patterns, we can access eight voxels for processing three voxel patterns simultaneously. That is to say, for processing a voxel pattern, the average number of accessed voxels would be $8/3 = 2.66$, which is fewer than that in the *MR* algorithm.

Theoretically, though we scan more rows, more voxel patterns are processed simultaneously, resulting in a lower average number of voxels to be accessed for processing a voxel pattern and thus a more efficient computing probably. On the other hand, with the increase of voxel patterns processed simultaneously, more and more states need to be considered. In the *ST* algorithm, we scan image rows in the raster scan and process voxel patterns one by one. There are $2^4 = 16$ states, and for each state, when we process a voxel pattern, four voxels need to be accessed; thus, we have to consider $2^4 \times 2^4 = 256$ cases. In the *MR* algorithm, we scan every two rows and process voxel patterns two by two. There are $2^6 = 64$ states, and for each state, we need to access six voxels; therefore, we need to consider $2^6 \times 2^6 = 4096$ cases. In the same way, if we scan every three rows and processed voxel patterns three by three, we need to consider $2^8 \times 2^8 = 65,536$ cases.

In general, when the interval of the rows to be scanned increases by 1, the voxel patterns to be processed simultaneously will increase by 1, and the number of considered cases will increase 16 times. As the number of cases to be considered increases, the complexity of the corresponding algorithm will also increase, and the algorithms may not be as efficient as expected in implementation. The experimental results presented in Section 4 demonstrate that although the *MR* algorithm performs more efficiently than the *ST* algorithm, it is not as efficient as expected.

For low density images, because there are rarely foreground voxels, for most of the voxel patterns in the image, the *CO* algorithm needs to access eight voxels in the voxel pattern being processed; therefore, it is less efficient than the *ST* algorithm and the *MR* algorithm. With the increased number of foreground voxels existing in the image, the efficiency of the *CO* algorithm will be improved gradually.

As mentioned above, if the voxel "*b*" in the voxel pattern is a foreground voxel, no other voxels need to be accessed for determining its Euler number increment. Thus, for processing a voxel pattern, we need to access one voxel only in this case. If the background voxels and the foreground voxels share the same probability of occurrence in the image, for half of the voxel patterns involved in the image, only one voxel needs to be accessed for determining the Euler number increments. Furthermore, we can determine 32 types of

voxel patterns' Euler number increments by accessing six voxels. Lastly, for the rest of the 96 types of voxel patterns, all the eight voxels need to be accessed for determining their Euler number increments. To sum up, for obtaining the Euler number of a 3D image with a resolution of $X \times Y \times Z$ voxels, the total voxel accessions will be $128/256 \times 1 \times X \times Y \times Z + 32/256 \times 6 \times X \times Y \times Z + 96/256 \times 8 \times X \times Y \times Z = 4.25 \times X \times Y \times Z$.

In practice, for the images with low densities of foreground voxels, the *CO* algorithm needs to access more voxels than those in the *ST* algorithm and the *MR* algorithm for processing a voxel pattern; thus, it is less efficient than the *ST* algorithm and the *MR* algorithm. On the other hand, for the images with moderate and high densities of foreground voxels, the number of accessed voxels for processing a voxel pattern in the *CO* algorithm is similar to or fewer than that in the *ST* algorithm and the *MR* algorithm. Thus, without extra transition among the states, the *CO* algorithm becomes the most efficient in the algorithms with our different strategies. The experimental results are consistent with our analysis.

## 6. Conclusions

For improving the voxel-pattern-based Euler number computing algorithm in 3D binary images, we introduced three strategies in this paper. The first strategy takes advantage of the information acquired during computing to avoid accessing voxels repeatedly and therefore can reduce the number of accessed voxels from 8 to 4 for determining the voxel pattern's Euler number increment. The second strategy involves scanning every two rows and processing voxel patterns two by two; therefore, only three voxels need to be accessed for determining the voxel pattern's Euler number increment. In the last strategy, the accessing order of voxels is changed when processing a voxel pattern in the image, and the voxel patterns with the same Euler number increments and consecutive indexes are combined into one group. Although this strategy can theoretically reduce the average number of accessed voxels for determining the Euler number increment from 8 to 4.25, it is more efficient than the previous strategies for moderate- and high-density images. Experimental results demonstrated that the three algorithms with each of our proposed three strategies respectively exhibit greater efficiency compared to the conventional voxel-pattern-based Euler number computing algorithm in most cases.

The voxels are processed in the raster scan order in the voxel-pattern-based Euler number computing algorithm; thus, these algorithms are suitable for parallel implementation. For our future work, we will consider hardware implementation and parallel implementation of the voxel-pattern-based Euler number computing algorithms on multiprocessors or FPGAs to further enhance the efficiency of Euler number computing.

**Author Contributions:** Methodology, B.Y. and L.H.; software, B.Y. and S.K.; validation, H.H. and Y.C.; writing—original draft preparation, B.Y.; writing—review and editing, L.H. All authors have read and agreed to the published version of the manuscript.

**Funding:** This research was funded in part by the National Natural Science Foundation of China under Grant Nos. 61971272 and 61603234, the Nitto Foundation, Japan, the Hibi Science Foundation, Japan and the Scientific Research Foundation of Shaanxi University of Science and Technology under Grant No. 2020BJ-18.

**Data Availability Statement:** The data presented in this study are available on request from the corresponding author.

**Conflicts of Interest:** The authors declare no conflict of interest.

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
