# Peer review of "Efficient Strategies for Computing Euler Number of a 3D Binary Image"

_electronics, doi:10.3390/electronics12071726_

Round 1

Reviewer 1 Report

The paper was well written.

Many mathematical models are present nowadays. Why did you choose the Euler number?

What is the advantage of your proposed method?

Check the all the mathematical formulas are correctly specified.

Check the reference section following the format

Check grammatical errors

Reviewer 2 Report

Efficient Strategies for Computing Euler Number of a 3D Binary 2 Image

The keywords should be of same format, either capitalaize first letter or remain small.

Author should improve background,motivation and contribution.

Author should discuss more reference in introducation setcion as below.

A)Well-balanced central schemes for the one and two-dimensional Euler systems with gravity

B)An overview of cluster-based image search result organization: background, techniques, and ongoing challenges

The authors must explain how their approach can handle the uncertainties.

Clearly highlight the mathematical terms used in the paper and explain them in the text.

Add border in all tables. And uniform the images position in whole paper

future work should be more focused at the end of conclusion.

The use of English language is fine, however, it is recommended to be checked once again.

Reviewer 3 Report

In this article, the authors proposed several strategies to efficiently reduce the calculation time for the Euler number from a 3D binary image. The methods description seems reasonable, and the new algorithms seem effective in terms of saving computation time. However, some descriptions and limitations of the algorithms need further clarification. Detailed comments are given below.

1.       The method seems only to be able to work with cubic voxels that only have binary values. Although this might be the dataset in three-dimensional image processing, some other examples include three-dimensional geometry defined by vertices and connection graphs. It is suggested to discuss when these three-dimensional binary data sets will be faced in real applications.

2.       In the demonstration, artificial gaussian noise patterns were used (Figure 7), which is different from real applications. It is suggested to test the algorithm on images acquired from experiments or other meaningful scenarios.

3.        It is suggested to explain the special abbreviation in the manuscript in the abstract or introduction, such as VP, ST, etc. The authors embed the explanation in the middle of the text, which makes it difficult to check the meaning of the terminologies.

4.       Reference for equation 2 (line 123) needs to be included or the full derivation should be included.

5.       In line 155, the authors wrote, “In other words, the increment of all the other 207 types of the voxel patterns is 0, thus, they will not affect the Euler number of a 3D image.” This should also be referenced unless the authors prove that this statement is correct.

Reviewer 4 Report

1) Lines 41-55 should be moved to the section the method is explained.

2) In the introduction, the problem should be mentioned concerning the literature and brief information about the study should be provided. The literature review is described in both Section 1 and Section 2. Relevant parts of these sections should be merged and if necessary should be given in a chapter of the literature review.

3) Section 3 should be simplified and shortened. A shorter explanation should be provided.

4) While the experimental results are mentioned, there is no detailed explanation of the experimental setup or methodology. It would be helpful for readers to understand how the experiments were conducted and what specific metrics were used to evaluate the efficiency of the proposed algorithms.

5) In the conclusion section, suggestions for future studies should be added.

Round 2

Reviewer 3 Report

The authors have addressed all my previous comments. I would like to recommend the publication of this manuscript.